# An Interdigital Microwave Sensor Based on Differential Structure for Dielectric Constant Characteristics Measurement

**DOI:** 10.3390/s23146551

**Published:** 2023-07-20

**Authors:** Xiaocong Tang, Zhiqiang Gao, Jie Wei, Zheyi Li, Yang Yi, Fan Yang, Azeem Muhammad, Cong Wang

**Affiliations:** 1School of Information and Communication, Harbin Institute of Technology, Harbin 150001, China; cbd0802@163.com (X.T.); weijie1996@stu.hit.edu.cn (J.W.); 21s005003@stu.hit.edu.cn (Z.L.); 22s105211@stu.hit.edu.cn (Y.Y.); yangfan13845850682@163.com (F.Y.); muhammad.azem03@gmail.com (A.M.); 2School of Astronautics, Harbin Institute of Technology, Harbin 150001, China

**Keywords:** microwave sensor, dielectric sensor, cross-finger structure, differential structure

## Abstract

In this work, a microwave resonator sensor with a unique configuration consisting of three resonators and two feedlines is proposed. This novel design aims to improve the performance and functionality of microwave resonator sensors for various applications. The frequency response of the sensor to materials with different dielectric constants is simulated. The results show that the most sensitive region of the sensor is located on the first interdigital structure, and placing the materials in other regions would enhance the linear correlation of its frequency response. The sensor also exhibits the ability to distinguish whether the same material has defects and the ability to qualitatively detect subtle changes in dielectric constant. Finally, the proposed sensor is fabricated and measured under the condition consistent with the simulation environment. The measured results are basically consistent with the simulation results, which confirms the potential of this sensor in detecting dielectric constants and resolving materials with defects, and the response of the sensor to the materials under test demonstrates its potential in measuring different thicknesses and loss tangents.

## 1. Introduction

Microwave resonator sensors, which use changes in a sample’s physical characteristics to gauge the resonant characteristics of a cavity or planar resonator, are a particular class of sensor. These sensors are widely applied to material characterization, biological sensing, environmental monitoring, etc. [1]. A sample is placed on the resonant cavity or on the resonator’s surface according to the fundamental concept of a microwave resonator sensor, and the resonant frequency or resonance amplitude of the cavity or resonator can be impacted by the changes in the sample’s physical characteristics and whether the sample is gaseous, liquid or solid [2,3,4]. The main types of microwave resonator sensor are cavity and planar [5]. Planar microwave sensors are now being researched by a number of groups for material characterization. At present, numerous RF resonant sensors using various configurations, such as the substrate-integrated waveguide (SIW) [6,7,8,9], split ring resonator (SRR), complementary split ring resonator [10,11,12,13], and the interdigital capacitors (IDC) [14,15,16,17,18]. The increased use of CSRR- and SRR-based microwave sensors among the several configurations mentioned above is due to their more compact designs and streamlined manufacturing processes. Early research in this field has also demonstrated that SRR-based structures and the metamaterial CSRR can provide a greater sensitivity than competing devices [19]. The maximal electric field of a planar microwave sensor based on the rectangular-shaped SRR is constrained in the SRR’s dielectric gap at resonance, providing a sensing spot for the analysis of dielectric materials. The shift in the resonant frequency is usually proportional to the change in the dielectric properties of the material, which is placed in the vicinity of the sensor. When a material with a high dielectric constant is placed near the resonator, it perturbs the electromagnetic fields around the resonator more than a material with a low dielectric constant. This perturbation causes a shift in the resonant frequency, which can be detected and measured [20].

Compared to other types of sensors, the interdigital capacitor-based microwave sensor provides a number of benefits. IDC sensors have the ability to identify subsurface structures, but they encounter a number of obstacles, including signal attenuation, dispersion, and interference from nearby objects [21]. Differential IDC sensors can solve these problems because they can eliminate common mode noise and interference, and produce more reliable and accurate measurement results [22]. In differential sensing, the difference between two signals is measured, as opposed to the absolute value of a single signal. This method can improve the sensor’s sensitivity and resolution, making it better suited to detecting subsurface structures [23].

In this paper, a microwave planar sensor is designed, simulated and fabricated; it presents a row of three parallel interdigital capacitors etched on the RF ground plane and a pair of feedlines is used to coupled and excite the sensor. A greater number of fingers are used for generating a high-intensity coupled resonating electric field in order to enhance its suitability for the sensitive measurement of the permittivity of solid materials and the frequency shifting when the material under test is placed. The sensor designed in this paper has a high sensitivity and fast response.

## 2. Design and Simulation of Microwave Sensor

The proposed interdigital microwave sensor was designed and fabricated using an F4B255 substrate with a relative dielectric constant of 2.55, a thickness of 1 mm and a loss tangent of 0.0015. The performance of the proposed sensor is simulated and optimized by the full wave electromagnetic software Computer Simulation Technology (CST), and the built-in time-domain solver of CST is used for simulation.

### 2.1. Structure of the Microwave Sensor

Firstly, an IDC differential sensor with three resonators is designed. The differential sensor with three resonators and a pair of electro-inductive transmission lines is shown below in Figure 1a. When the lines are loaded in an asymmetric manner, the differential sensor operates based on mode conversion. In other words, if the lines are not loaded or if both lines are loaded with the same materials in their sensitive areas, mode conversion does not occur, and ideally, the cross-mode transmission coefficient is zero. The results of simulated S-parameters are shown in Figure 1b. From the S-parameter diagram of the sensor, it can be seen that when the frequency is around 1.5 GHz, the properties of the sensor are closer to the resonant circuit. There is a resonance peak in S_11_ parameter, and a considerable portion of energy is transmitted to Port 2 in S_21_ parameter; when the frequency is about 5.3 GHz, the sensor is similar to the antenna in nature. The S_11_ parameter has an obvious transmission zero, and the S_21_ parameter shows that almost all the energy is radiated into space. When using this device as a sensor, we prefer it to have the characteristics of a resonant circuit. The results show that the resonance amplitude and Q-factor of the sensor are very low, at 1.5 GHz, and optimization is needed to improve the resonance amplitude and Q-factor.

After simulation, it is found that, by adjusting the values of devices W1, W2, and L1, the resonant characteristics of the device can be changed, thereby achieving the effect of optimizing device performance. Figure 2a–c show the impact of changing W1, W2, and L1 on device performance. W1 has little effect on the parameters of device S_11_. When W1 = 1.6 mm, reducing the size of W1 will almost not affect the parameters of the S_11_. The device performance is optimal when W2 = 1 mm. Increasing or decreasing the value of W2 will result in a decrease in the Q-factor of the device. The increase in L1 value will cause an increase in Q-factor, but due to the limitations of the overall size of the device, the maximum value of L1 can only reach 4 mm.

The two-dimensional model diagram of the optimized sensor is shown in Figure 3a; Figure 3b shows the simulated S-parameters, and Figure 3c shows the electric field distribution on the sensor. Table 1 records its specific size parameters. The results show that the resonant frequency of the device is 1.065 GHz. From the S-parameters, it can be seen that the optimized sensor has obvious resonance circuit characteristics, which meets the requirements for use as a sensor. The three resonators that make up the sensor correspond to Resonator 1st (R1), Resonator 2nd (R2), and Resonator 3rd (R3). From the electric field distribution, it can be seen that the electric field at R1 is the strongest, and R1 is most likely the most sensitive area of the device.

### 2.2. Simulation Response of the Microwave Sensor

In order to verify the response of the sensor to the dielectric constant of the material being measured and determine the location of the sensitive area, a simulation analysis is conducted. The input port is set as the feeder port of the R1 region, and the output port is set as the feeder port of the R3 region. Different materials under test (MUTs) are placed on different resonators and five types of dielectric materials are used in the analysis, namely F4BM, F4BTM, F4BTM1, TP, and TP1. Their loss tangent is 0.001, their thickness is 1.5 mm, and their dielectric constants are 2.2, 4.0, 6.15, 9.6, and 10.5, respectively. These MUTs are placed on the resonator of the sensor with seven different cases; different MUTs are placed on R1 (As shown in Figure 4) and compared with the result without MUTs. Similarly, a different MUT is placed on R1 and R3, R1&R2, R1&R2&R3, R1&R3, R2&R3, and the results are compared to determine how much the magnitude response has shifted. The simulated response of sensor S-parameters is shown in Figure 5a–g. In order to more intuitively demonstrate the impact of MUT placement area on sensor response, the frequency deviation in sensor resonance frequency caused by different MUTs is plotted in Figure 5h, and the linear correlation coefficients (R^2^) of the curve are calculated and shown in Table 2. The simulation results are consistent with the expected assumption that R1 is the most sensitive area of the sensor. When an MUT is placed in R1, the sensor has a great response in frequency; when the MUT is placed in other areas, the sensor response is extremely poor and does not have good linearity. This is consistent with the simulation results of the distribution of electric field intensity on the surface of the sensor. As it has the highest field strength intensity among these resonators, R1 is the most sensitive area of the sensor, while the R2&R3 area has an almost poor response to the MUT because of its low electric field intensity. This method is equivalent to placing MUTs simultaneously in the symmetrical area of two lines, so there will be no mode switching and the response of the sensor will not change significantly. In addition, the results also show that placing the MUT in R1 while also placing the material in other areas can improve the linear correlation coefficient of frequency response and enhance the linearity of the sensor. The reason for this phenomenon may be that although the field strength in the R2&R3 region is very low, the frequency response of the sensor is poor when the material to be tested is placed separately above it; however, when they are placed on the material to be tested together with R1, this can enhance the overall impact of the sensor.

On this basis, the difference in frequency response is simulated and analyzed between the tested material with defects and the uniformly tested material. In actual measurements, the MUT probably contain defects such as voids, which can result in its equivalent dielectric constant being slightly lower than that of a homogeneous material. In order to test the response of sensors to subtle changes in the dielectric constant of the tested material and simulate measurement scenarios, where the tested material has defects, three holes with a diameter of 0.6 mm and a spacing of 4 mm are set up on the simulated material model, all of which need to be distributed on the axis of the tested material. Similarly, these materials with holes are placed in different regions. If they are placed in the R1 region (as shown in Figure 6), their corresponding simulation results are marked as DR1, and so on. Figure 7 shows the S_11_ parameters of sensor with two materials placed in different areas, and the corresponding frequency offsets are summarized in Table 3. The results show that when the MUT is placed in the R1 and R1&R3 regions, the sensor shows the most significant response to subtle dielectric constant changes, indicating its potential in defect detection of the tested material. Interestingly, when the MUT is placed in R1 and R2, the response of the sensor disappears. It is speculated that placing the MUT alone in R2 will cause the resonance frequency of the sensor to shift to the right or split, thereby maintaining the overall resonance frequency of the sensor.

## 3. Results and Discussion

In order to verify the reliability of the simulation results, a sensor that is consistent with the simulation model is fabricated and its frequency response to materials with different dielectric constants is tested. The vector network analyzer (VNA) used in the test is Keysight’s N9916A. After the measurement frequency range is set to 0.5–2 GHz, the 85521A calibrator is used for dual-port calibration. After calibration, Port 1 of VNA is connected to the input terminal of the sensor, and Port 2 is connected to the output terminal of the sensor for testing. A measurement setup using a differential microwave interdigital sensor with three resonators is employed. The objective is to measure the response of the different MUTs placed on each resonator and analyze the impact of their dielectric properties on the sensor’s performance. Five different MUTs are utilized in the measurement process: F4BM, F4BTM, F4BTM1, TP, and TP1. Their loss tangent is 0.001, their thickness is 1.5 mm, and their dielectric constants are 2.2, 4.0, 6.15, 9.6, and 10.5, which are consistent with the parameters of the simulation model. In order to avoid the impact of fixtures on test results, no fixtures are used during the testing process, and MUTs are directly placed on the sensors. The actual testing environment and the physical object of the sensor are shown in Figure 8a. When no material is placed on the sensor for testing, the test result of S_11_ parameter is shown in Figure 8b. This shows that its resonant frequency is 1.2275 GHz, basically consistent with the simulation results, and the placement of MUTs during testing is shown in Figure 9. To match the simulation model, MUTs are cut into appropriate sizes, with each MUT covering a separated area.

Figure 10a–g shows the S_11_ parameters of sensors with different dielectric constants placed in different regions, and summarizes the frequency shift response of the sensor in Figure 10h. The measured results are basically consistent with the simulation results. When the tested material is placed in R1, R1&R2, R1&R3, R1&R2&R3, the sensor exhibits a clear and regular response; when the MUT is placed in R2, R3, and R2&R3, the response of the sensor is poor, and there is no obvious pattern of frequency offset. The R^2^ of the frequency response curve is summarized in Table 4. Since the resonance frequency does not shift regularly when the material being tested is placed on R2 or R3, R^2^ cannot be calculated. The results show that placing the material to be tested on R1 while also placing it on R2 or R3 can significantly improve the linearity of the response curve, which is in good agreement with the simulation results. Normalized sensitivity (S) is also defined here, presented as a percentage, and calculated as follows:S = Δ*f*/f_r_ × 100
where Δ*f* is the frequency offset caused by changes in the unit dielectric constant, and f_r_ is the original resonant frequency point of the sensor. After calculation, the S of this sensor is 0.93%.

S_22_ parameters of the sensor are used to analyze the differential mode response of the sensor. Figure 11 shows the results when MUTs with different dielectric constants are placed at R1. When the dielectric constants of MUTs are 2.2 and 4.0, there is almost no change in the lower-resonance frequency, while the higher-resonance frequency has a significant shift. When the dielectric constants of MUTs are 6.15, 9.6 and 10.5, the lower-resonance frequency shifts significantly, while the higher-resonance frequency has no change. This phenomenon can be attributed to the insertion of an additional interdigital structure between the output and input ports, which improves the linearity of the sensor and affects the coupling of the two feeder lines, causing resonance frequency splitting in S_22_ parameter and resulting in the differential mode response of the sensor deviating from the ideal state. However, after a detailed analysis of the split resonant frequency as a reference resonant frequency point, the differential mode response of the sensor can still be used to eliminate the external interference.

In addition, alumina ceramic and soda lime glass are also detected as MUTs. Figure 12 shows the S_11_ parameters corresponding to the sensor when they are placed in different regions, and the extracted frequency shift is shown in Table 5. The dielectric constant of alumina ceramics is 9.4, the loss tangent is 0.0003, and the thickness of the sample is 0.5 mm; the dielectric constant of calcium sodium glass is 8.3, the loss tangent is 0.0001, and the thickness of the sample is 1.5 mm. The results show that, when placing calcium sodium glass with lower dielectric constant on the sensor, the frequency shift value is higher, which can be attributed to the larger thickness of the calcium sodium glass, demonstrating the sensor’s potential in sample thickness detection. On the other hand, the dielectric constant of the dielectric material TP is 9.6, which is similar to that of alumina ceramics. The sensor has a greater frequency response to alumina ceramics, which can be attributed to the lower dielectric loss of alumina ceramics, which demonstrates the potential of the sensor in detecting the loss tangent.

The response of the sensor to defective materials is also measured, and a comparison between the obtained results and the response of the sensor to non-defective materials is shown in Figure 13. The frequency shift extracted from this is shown in Table 6. The results show that when the MUT is placed on R1, R1&R3, R1&R2&R3, the frequency shift response of the sensor reaches its maximum value of 7.5 MHz. This is basically consistent with the simulation results, demonstrating the potential of the sensor in the defect detection of the tested MUTs.

On this basis, three types of MUTs with different defect distributions are detected by sensor, and the results are shown in Figure 14. During the test, MUTs are placed in the R1 region. To ensure consistency in testing, the sensor will first measure the response of the intact MUTs, and then measure the sensor’s response to the punch holes in MUTs. Compared to the intact MUTs, the frequency shift of the sensor’s response to MUTs with defects is summarized in Table 7. The results show that if the MUT has the same number of defect voids, the frequency shift will basically be the same, indicating that the detection of defects by the sensor is qualitative. A comparison with the reported resonant sensors is summarized in Table 8, and it can be seen that the sensor proposed in this work has the best overall sensing performance.

## 4. Conclusions

This paper presents an implementation strategy for a microwave interdigital sensor with high sensitivity and a quick response time. The simplicity of its design and its compatibility with other microwave components make it a highly versatile tool with a wide range of potential uses. Through various experiments involving different interdigital figure arrangements and gap placements, a final sensor shape based on the favorable characteristics observed in S_11_ and S_21_ measurements was selected, which yielded favorable results. Material testing was also performed to validate the sensor’s effectiveness across different scenarios The simulation results indicate a resonant frequency of 1.064 GHz. When the material under testing is placed in R1, R1&R2, R1&R3, R1&R2&R3, the sensor exhibits a significant frequency shift response and demonstrates the ability to distinguish qualitative sample defects. The measured frequency offset response of the sensor closely aligns with the simulated results. When the tested material is placed on R1, R1&R2, R1&R3, R1&R2&R3, the sensor demonstrates a substantial frequency-offset response. For the material to be tested in the borehole, the measured results exhibit a frequency offset response of up to 7.5 MHz, indicating the sensor’s potential for dielectric constant detection and material defect detection. Additionally, the sensor’s response to MUT with varying thicknesses and loss tangents shows its potential for detecting thickness and loss tangent variations.

## Figures and Tables

**Figure 1 sensors-23-06551-f001:**
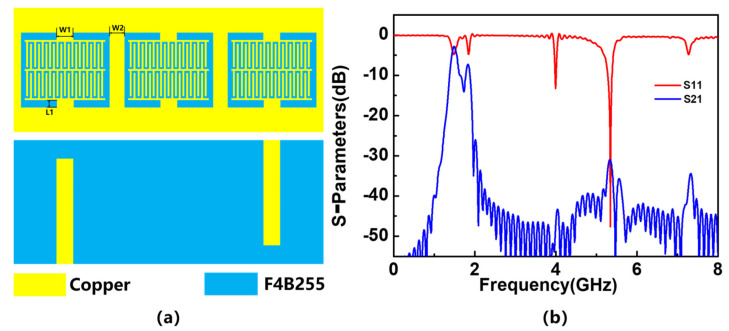
2D model of IDC differential sensor and simulated S-parameters: (**a**) model and (**b**) S-parameters.

**Figure 2 sensors-23-06551-f002:**
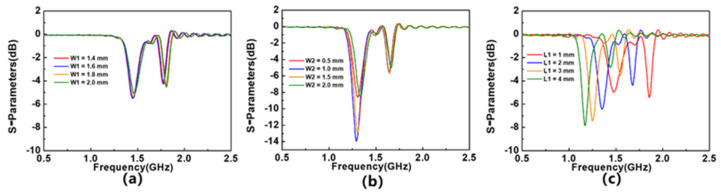
The influence of (**a**) W1, (**b**) W2, and (**c**) L1 value on device S_11_ parameters.

**Figure 3 sensors-23-06551-f003:**
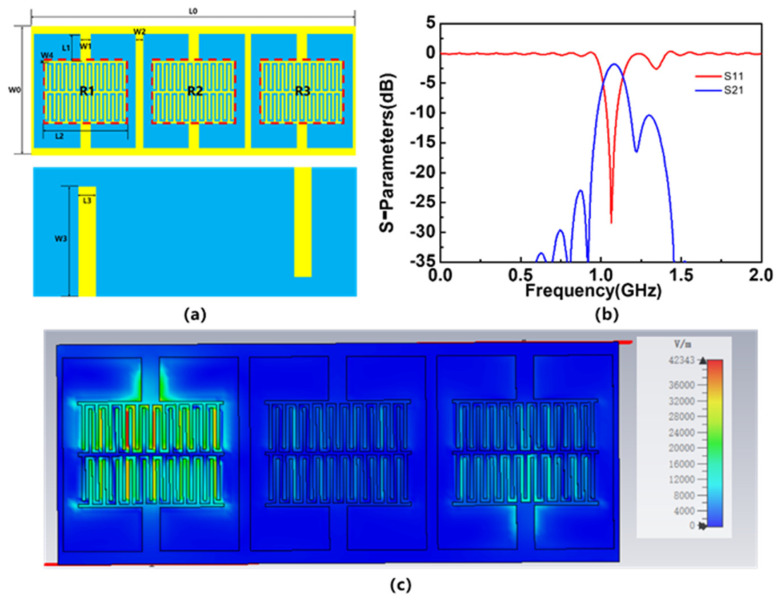
2D model of optimized sensor, the simulated S-parameters of the sensor and electric field distribution of the optimized sensor: (**a**) model, (**b**) S-parameters, and (**c**) electric field distribution.

**Figure 4 sensors-23-06551-f004:**
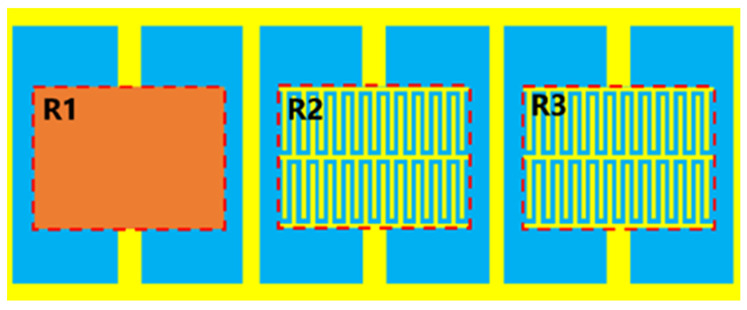
The material to be tested is placed on R1.

**Figure 5 sensors-23-06551-f005:**
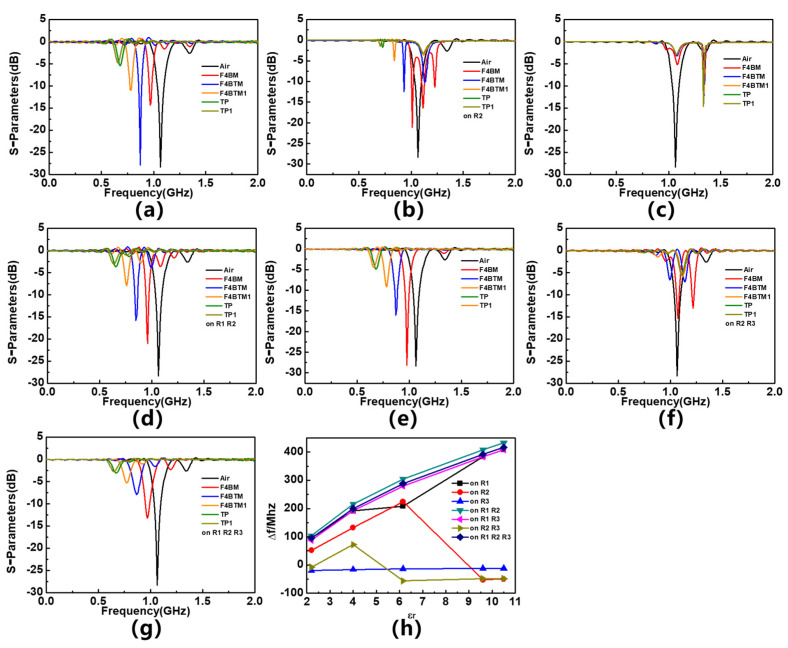
Simulated S_11_ parameters of different MUTs placed (**a**) on 1st resonator, (**b**) on 2nd resonator, (**c**) on 3rd resonator, (**d**) on 1st and 2nd resonators, (**e**) on 1st and 3rd resonators, (**f**) on 2nd and 3rd resonators, (**g**) on 1st, 2nd and 3rd resonator, and (**h**) frequency offset response of the differential sensor.

**Figure 6 sensors-23-06551-f006:**
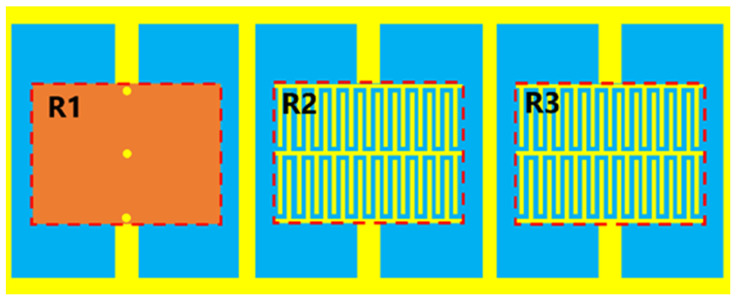
The material with holes is placed in R1.

**Figure 7 sensors-23-06551-f007:**
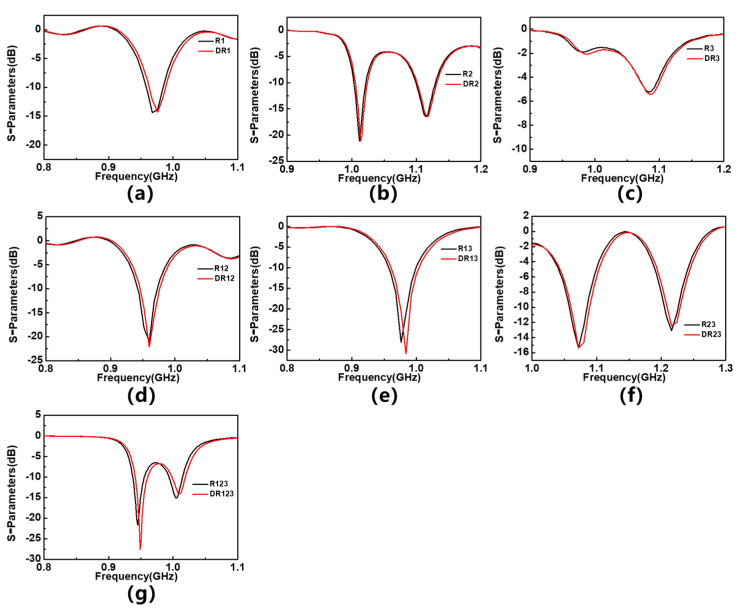
The frequency offset response of F4BM with and without holes (**a**) on 1st resonator, (**b**) on 2nd resonator, (**c**) on 3rd resonator, (**d**) on 1st and 2nd resonators, (**e**) on 1st and 3rd resonators, (**f**) on 2nd and 3rd resonators, and (**g**) on 1st, 2nd and 3rd resonators.

**Figure 8 sensors-23-06551-f008:**
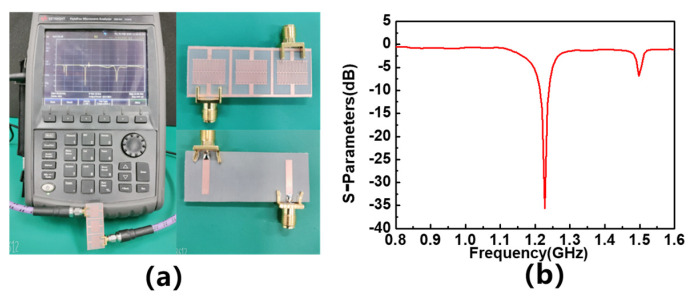
Experimental setup to evaluate the performance of the sensor and the measurement result of S_11_ parameter: (**a**) testing environment and (**b**) S_11_ parameter.

**Figure 9 sensors-23-06551-f009:**
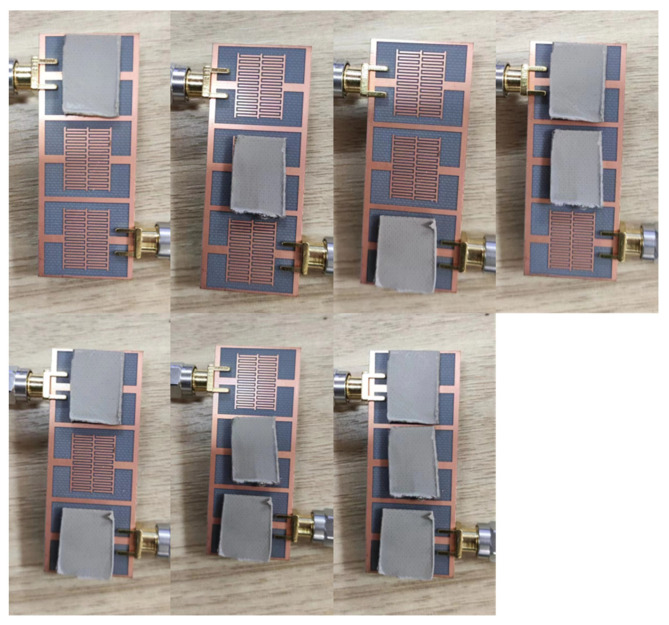
The placement of the MUTs.

**Figure 10 sensors-23-06551-f010:**
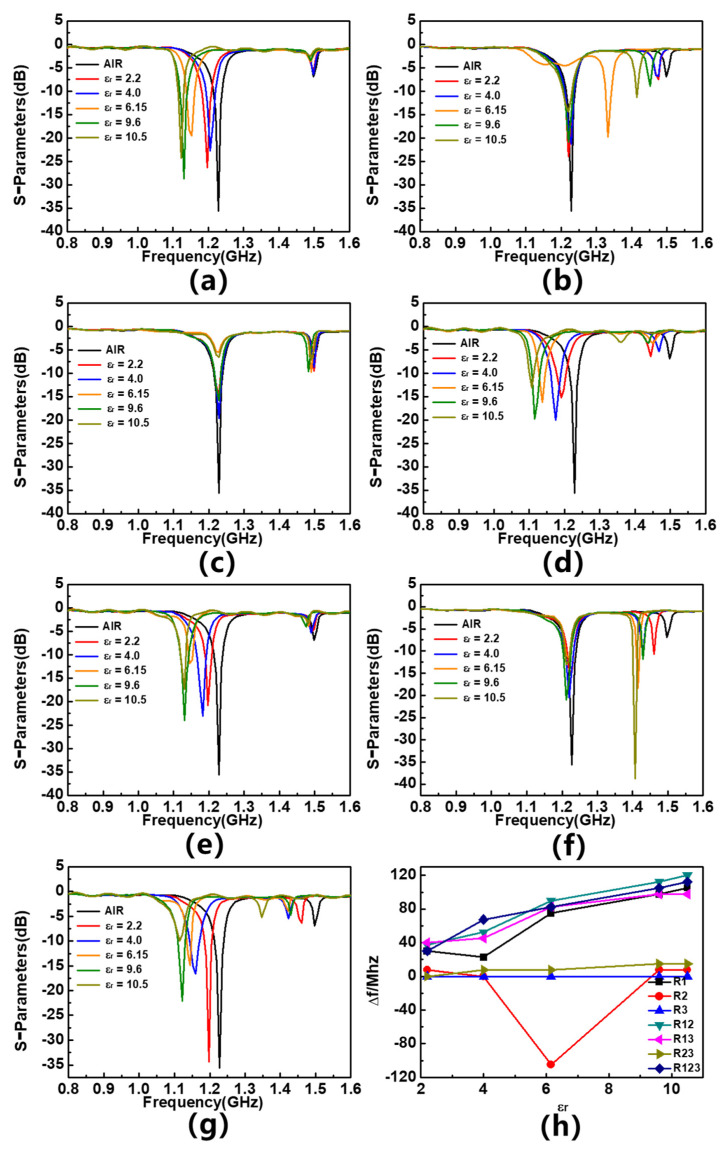
S_11_ parameters of different MUTs placed on (**a**) 1st resonator, (**b**) 2nd resonator, (**c**) 3rd resonator, (**d**) 1st and 2nd resonators, (**e**); 1st and 3rd resonators, (**f**) 2nd and 3rd resonators, (**g**) 1st, 2nd and 3rd resonators, and (**h**) frequency offset response of sensor.

**Figure 11 sensors-23-06551-f011:**
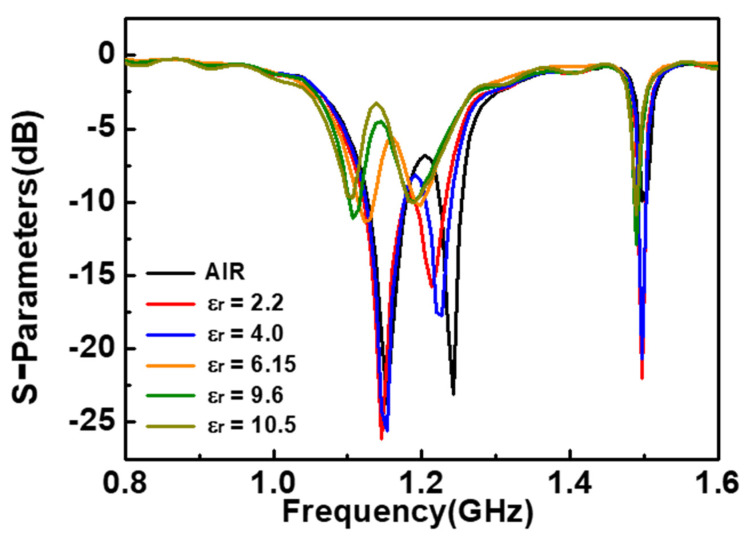
S_22_ parameters of different MUTs placed on 1st resonator.

**Figure 12 sensors-23-06551-f012:**
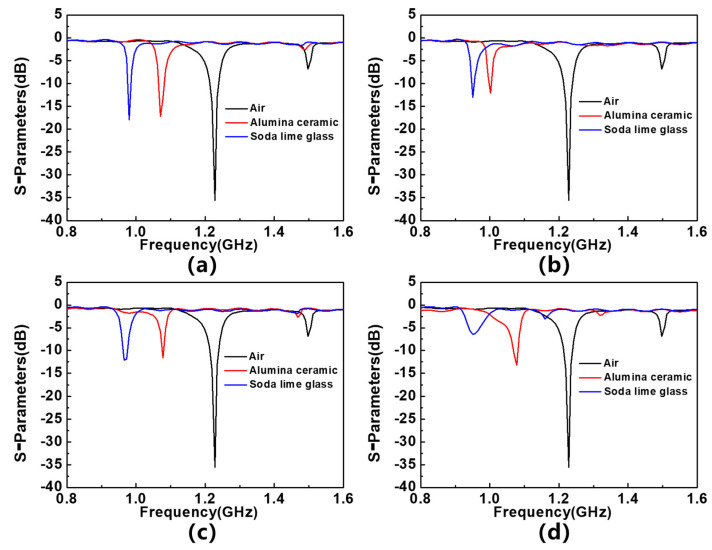
S_11_ parameters of alumina ceramic and soda lime glass placed on (**a**) 1st resonator, (**b**) 1st and 2nd resonators, (**c**) 1st and 3rd resonators, and (**d**) 1st, 2nd and 3rd resonators.

**Figure 13 sensors-23-06551-f013:**
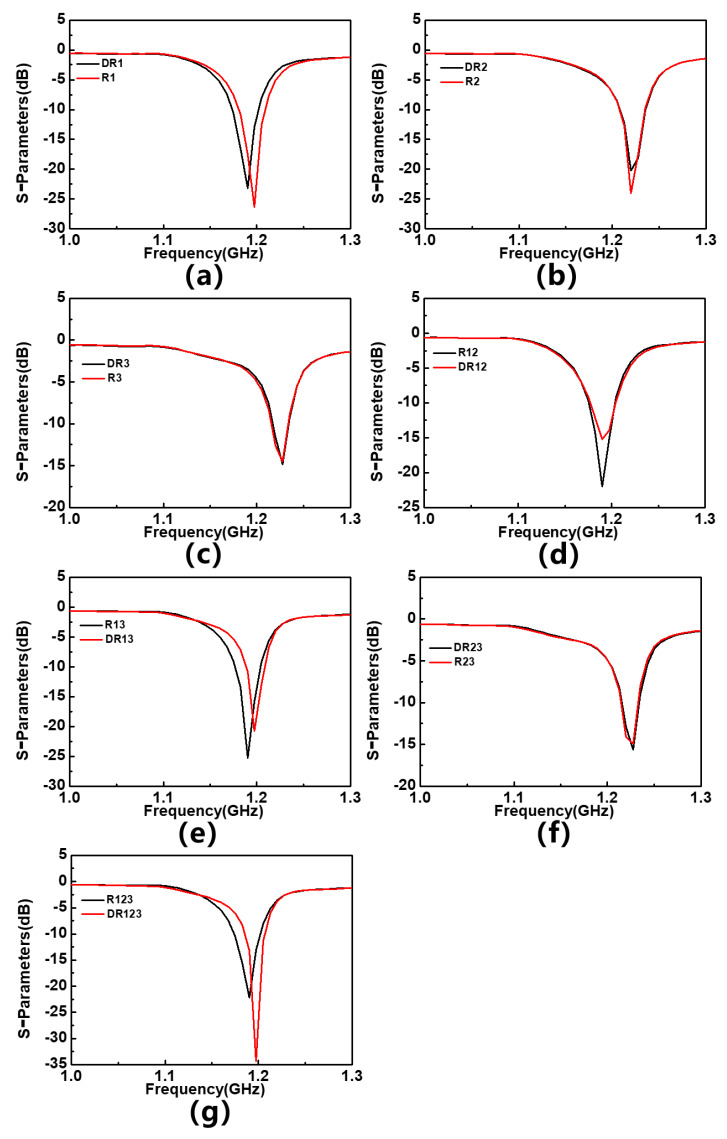
S_11_ parameters of different defective MUTs placed on (**a**) 1st resonator, (**b**) 2nd resonator, (**c**) 3rd resonator, (**d**) 1st and 2nd resonators, (**e**) 1st and 3rd resonators, (**f**) 2nd and 3rd resonators, and (**g**) 1st, 2nd and 3rd resonators.

**Figure 14 sensors-23-06551-f014:**
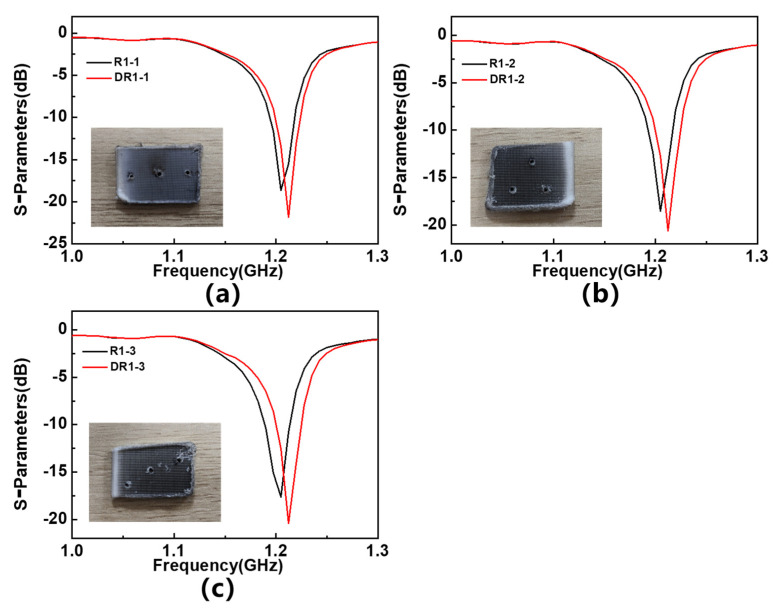
S_11_ parameters of MUTs with different defect distributions: (**a**) central axis distribution, (**b**) triangular distribution, and (**c**) diagonal distribution.

**Table 1 sensors-23-06551-t001:** Dimensional parameters of the proposed sensor.

**Name**	W0	L0	W1	L1	W2	L2	W3	L3	W4
**Value (mm)**	20.0	50.1	1.6	4.0	1.0	13.0	17.0	2.7	0.3

**Table 2 sensors-23-06551-t002:** Simulated linear correlation coefficients of frequency offset response without defects.

**Zone**	R1	R2	R3	R12	R13	R23	R123
**R^2^**	0.959	0.123	0.8220	0.967	0.975	0.172	0.975

**Table 3 sensors-23-06551-t003:** Simulated frequency shift response after placing defected MUTs.

**Zone**	R1	R2	R3	R12	R13	R23	R123
**Δ*f* (MHz)**	8	0	0	0	8	0	5

**Table 4 sensors-23-06551-t004:** Measured linear correlation coefficients of frequency offset response without defects.

**Zone**	R1	R2	R3	R12	R13	R23	R123
**R^2^**	0.874	—	—	0.960	0.889	0.895	0.914

**Table 5 sensors-23-06551-t005:** Measured frequency shift response after placing alumina ceramic and soda lime glass.

**Zone**		R1	R12	R13	R123
**Δ*f* (MHz)**	alumina ceramic	157.5	112.5	150	150
	soda lime glass	247.5	277.5	262.5	270

**Table 6 sensors-23-06551-t006:** Measured frequency shift response after placing defected MUTs.

**Zone**	R1	R2	R3	R12	R13	R23	R123
**Δ*f* (MHz)**	7.5	0	0	0	7.5	0	7.5

**Table 7 sensors-23-06551-t007:** Measured frequency shift response after placing an MUT with different defect distributions.

**Defect Distribution**	Central axis	Triangle	Diagonal
**Δ*f* (MHz)**	12	12	12

**Table 8 sensors-23-06551-t008:** Comparison table with the previous works.

Reference	S (%)	R^2^
[24]	0.26	—
[25]	0.59	—
[26]	0.41	0.99
[27]	0.03–0.07	—
This work	0.93	0.96

## Data Availability

The data presented in this study are available on request from the corresponding author. The data are not publicly available due to it will be used for subsequent research.

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
