# Peer review of "An Interdigital Microwave Sensor Based on Differential Structure for Dielectric Constant Characteristics Measurement"

_sensors, 2023, doi:10.3390/s23146551_

Round 1

Reviewer 1 Report

An Interdigital Microwave Sensor Base on Differential Structure for Dielectric Constant Characteristics Measurement

This paper presents a reflection-mode microwave resonator sensor with capacitive regions that are implemented using an IDC topology. The performance of the sensor is well studied and the results are in good agreement with the simulations. However, there are a few points regarding the presentation of the results that are needed to be addressed. The following are discussing the details of my comment:

Comment 1:

The dielectric properties of sample materials (MUTs) are not clearly presented in the paper. I would also recommend having photographs of the samples on the sensor to better visualize the sensitivity and performance of the sensor and the three constituent resonators.

Comment 2:

There are several related works on the development of microwave sensors based on SRRs with single and differential topologies, that the paper can include for a comperehensive literature review:

10.1109/TMTT.2021.3109599

10.1109/TMTT.2022.3142038

10.1109/JSEN.2020.3041810

Comment 3:

Figure 9 and 10 represent the most significant results of the paper, yet their quality is quite poor. Please revise the figures to match the font sizes and legibility.

Comment 4:

The language of the paper can be revised to avoid informal and uncommon phrases in scientific writing. Here are two instances from the paper:

Obviously, the resonant circuit characteristics of this device are not satisfactory, so further optimization is needed based on it.

Many applications, such as material characterization, chemical and biological sensing, use these sensors extensive.

Comment 5:

The abstract needs to be revised to include quantitative performance measures of the sensor. Statements such as: The sensor also exhibits the ability to distinguish between materials and subtle changes in dielectric constant are solely qualitative and can’t navigate the reader on the performance assessment of the sensor.

good

Reviewer 2 Report

The manuscript proposed an interdigital microwave sensor base on differential structure. The sensor exhibits the ability to distinguish between materials and subtle changes in dielectric constant. The sensor has potential for permittivity detection and defective material discrimination. It is recommended for publication after addressing these issues.

1.Which color in Figure 1(a) represents F4B255, blue or yellow? What does the other color represent?

2. Which method is used in simulation, time domain solver or frequency domain solver?

3. In Figure 1(b), What causes the other resonant peaks except for the peaks at 1.5 GHz and 5.3 GHz? What are the subsequent optimization criteria?

4. How did the test placed the samples? two or three samples placed in different areas at the same time or one sample cover two or three areas? Does it affect the test results? Suggest adding test photos.

5. It is recommended to do more comparison experiments with several groups of random distribution of defects.

6. It is recommended to list the test results of several reference material, such as PTFE, fused silica, alumina, etc.

Minor editing of English language required

Reviewer 3 Report

The authors present a sensor for determining the permittivity of a sample using an interdigital microstrip structure claimed to operate in differential mode. In this regard, the interdigital structure is widely used in microwave resonators, as well as the application and collection of power waves in differential form to obtain the reflection coefficient of structures. So, the authors are not highlighting any novelty in their submission. Besides, recent important references on the topic are missing, for instance:

N. Zhang, et al, “Dual-mode anti-interference humidity detection: Differential microwave sensor based on microstrip circuit” Sensors and Actuators B: Chemical, Volume 390, 1 September 2023.

Yeo, J.; Lee, J.-I. High-Sensitivity Microwave Sensor Based on an Interdigital-Capacitor-Shaped Defected Ground Structure for Permittivity Characterization. Sensors 2019, 19, 498.

Some additional comments:

Based on the large number of structures to provide results in the same bandwidth, additional features such as extraction of the material loss tangent should complement the proposal.

The differential-mode response of the sensor is not explained; input and output ports are not indicated.

The measurement setup is not detailed, whereas the description of the calibration and fixture de-embedding is not explained.

Even though the paper is understandable, it should be written in a more concise manner; particularly the introduction section. On the other hand,  the description of figures within the body text should include relevant information concluded from them.

Round 2

Reviewer 3 Report

The authors perform measurements using their structure. Then, they correlate an EM model with experimental data to show that, through the parameter extraction they made, it is possible to reproduce the S-parameters of their structure when there exists interaction with a sample of certain characteristics. This shows no evidence that their structure, at least in its current form, is useful as an interface to measure the complex permittivity of materials (i.e., epsilon’ and epsilon’’). Therefore, I would consider recommending this paper for publication in case the authors either include evidence that the relative permittivity and loss tangent of materials have accurately been determined. Alternatively, the abstract, introduction and conclusion sections should be modified to explain that their structure is sensitive to changes in the permittivity of measured samples, but further experiments are necessary to demonstrate the potential to accurately determining the complex epsilon.

There are many typo errors throughout the manuscript, starting from the title.
